# eDNA metabarcoding as a biomonitoring tool for marine protected areas

**Zachary Gold**[1]*, **Joshua Sprague**[2], **David J. Kushner**[2], **Erick Zerecero Marin**[1], **Paul H. Barber**[1]

**1** Department of Ecology and Evolutionary Biology, University of California–Los Angeles, Los Angeles, California, United States of America, **2** Channel Islands National Park Service, Ventura, California, United States of America

* zack.gold@ucla.edu

**Data Availability Statement:** All data generated for this study are available through Dryad as the FishCARD link (https://doi.org/10.5068/D1H963)

## Abstract

Monitoring of marine protected areas (MPAs) is critical for marine ecosystem management, yet current protocols rely on SCUBA-based visual surveys that are costly and time consuming, limiting their scope and effectiveness. Environmental DNA (eDNA) metabarcoding is a promising alternative for marine ecosystem monitoring, but more direct comparisons to visual surveys are needed to understand the strengths and limitations of each approach. This study compares fish communities inside and outside the Scorpion State Marine Reserve off Santa Cruz Island, CA using eDNA metabarcoding and underwater visual census surveys. Results from eDNA captured 76% (19/25) of fish species and 95% (19/20) of fish genera observed during pairwise underwater visual census. Species missed by eDNA were due to the inability of MiFish *12S* barcodes to differentiate species of rockfishes (*Sebastes*, n = 4) or low site occupancy rates of crevice-dwelling *Lythrypnus* gobies. However, eDNA detected an additional 23 fish species not recorded in paired visual surveys, but previously reported from prior visual surveys, highlighting the sensitivity of eDNA. Significant variation in eDNA signatures by location (50 m) and site (~1000 m) demonstrates the sensitivity of eDNA to address key questions such as community composition inside and outside MPAs. Results demonstrate the utility of eDNA metabarcoding for monitoring marine ecosystems, providing an important complementary tool to visual methods.

## Introduction

Marine Protected Areas (MPAs) promote sustainability of marine ecosystems and the ecological goods and services they provide [1]. However, ensuring MPA effectiveness requires regular monitoring to document that ecosystem health is stable or improving [1]. MPA monitoring also provides an essential opportunity to assess the impact of management practices, allowing resource managers to adjust management plans as required [2].

Current MPA monitoring protocols typically assess the diversity and abundance of fish and benthic invertebrates, as well as community trophic structure SummaryTable. Much of this assessment is based on underwater visual census surveys conducted on SCUBA [3], which are

and the eDNA MPA data link (https://doi.org/10.5068/D1SQ3K). All code are available on GitHub (https://github.com/zjgold/FishCARD). A Zenodo link is linked to the version of the FishCard code (https://doi.org/10.5281/zenodo.4315278). Generated sequences are available on NCBI SRA (https://www.ncbi.nlm.nih.gov/sra/PRJNA681428).

**Funding:** National Geographic Young Explorer Grant (992916); Catalyst Grant: UC Conservation Genomics Consortium (CA-16-376437); and Resources Legacy Fund Foundation (12481) funded this research. Z.G. and E.Z. were supported by the US-NSF Graduate Research Fellowship, grant number DEG No. 1650604.

**Competing interests:** The authors have declared that no competing interests exist.

costly, and time and labor intensive [3]. For example, to survey 33 sites within the Channel Islands National Park once per year, the National Park Service Kelp Forest Monitoring Program [4] spends approximately ~1,000 hours of dive time and ~1,400 hours performing data entry, data checking and quality assurance/quality control [59]. Furthermore, SCUBA-based surveys are constrained by weather, diving conditions, and personnel [59], and can require extended and repeated dives to accurately document marine communities that place divers at risk for dive-related injuries. SCUBA surveys can also introduce significant observer bias, as fish react differently to divers, particularly inside and outside of MPAs, potentially impacting survey results [5].

Given the above logistical and methodological constraints, MPA monitoring efforts are largely limited to shallow depths (e.g. <30 m) and the most economically or ecologically important taxa as proxies for ecosystem health [6]. Moreover, examining a predetermined subset of community diversity potentially excludes crucial functional groups, biasing ecosystem assessment [7]. Combined, these issues restrict the scope, scale, and frequency of visual surveys, limiting the utility of SCUBA-based MPA surveys to quantify biodiversity and trophic structure [3], data essential for assessing MPA effectiveness.

One promising new approach for assessing and monitoring marine ecosystems is environmental DNA, or "eDNA", a technique based on isolation and sequencing of freely associated DNA from soil or water samples [8]. Through metabarcoding and high-throughput next generation sequencing, eDNA can broadly survey community biodiversity in a rapid, repeatable, and affordable manner [8]. As such, eDNA is ideally suited to measure the biodiversity for intensive monitoring programs, such as those required for MPAs [9].

eDNA has some key advantages over traditional SCUBA-based survey methods for biodiversity measurements. First, eDNA can capture a wide diversity of marine vertebrate taxa, frequently detecting more species than traditional fish survey methods [10]. Second, eDNA detects rare and cryptic species that are frequently overlooked or ignored in traditional survey methods [11,12], including both endangered and invasive species [8]. Third, eDNA collection is relatively simple, requiring only small volumes of seawater (e.g. < 3 L) and simple filtering techniques, allowing sampling by individuals with limited training, even in remote locations [13]. Forth, because eDNA doesn't require diving, there are significant worker safety advantages. Lastly, eDNA is affordable (e.g. ~$50/sample) and has the potential for automation, allowing for remote sample collection and high throughput autonomous lab processing [14].

Despite these advantages, eDNA also has limitations. Of particular concern is PCR bias that can result in preferential amplification of particular taxa [15]. Additionally, detection probabilities can be influenced by species specific eDNA generation and degradation rates [8], an issue potentially further complicated by the transport of eDNA on ocean currents [16]. Furthermore, primer design, bioinformatic, and reference database limitations can also affect the accuracy of taxonomic assignment from eDNA [17].

Unlike well-established visual surveys, the impact of biases in eDNA metabarcoding are not well characterized, and may be less problematic than believed. For example recent studies show that impacts of PCR bias can be mitigated by technical replicates and site occupancy modelling [17–19]. Similarly, because eDNA signals decay relatively rapidly (e.g. hours to days; [20,21]), eDNA signatures are surprisingly stable [22]. As such, eDNA holds tremendous promise for monitoring marine ecosystems. Realizing that promise, however, requires a better understanding of how visual surveys and eDNA metabarcoding approaches compare in direct field applications.

The Channel Islands MPA Network spans >1000 reefs across six islands off the coast of Southern California. It is monitored by several programs including the Kelp Forest Monitoring Program, which conducts visual monitoring surveys of 41 invertebrates and over 100 fishes

[4]. In total only 94 of the >1000 Channel Island reefs are surveyed, and just once per year [6], missing the seasonal dynamics in the variable Southern California Bight, limiting the scope and scale of assessment [2]. While born of logistical necessity, the spatial and temporal limits of this survey protocol makes accurately assessing the health of this MPA network difficult [2,23] and suggests the need for new approaches that produce data on broader taxonomic, spatial and temporal scales.

This study tests the efficacy of eDNA for MPA monitoring and to better understand the advantages and shortcomings of eDNA methods. We do this through a side-by-side comparisons of eDNA metabarcoding and visual surveys of fish communities conducted by the National Park Service.

## Materials and methods

### Sample collection

We conducted our study at Scorpion State Marine Reserve within the Channel Islands National Park and National Marine Sanctuary under the State of California's Natural Resources Agency Department of Fish and Game scientific collection permit

(SCP) number: 13898. To determine the degree to which eDNA could capture documented differences inside and outside this MPA, we sampled three sites: 1) inside the MPA (34.05223 N, 119.58253 W) 2) outside but adjacent (<0.5km) to the MPA ("edge site"; 34.04415 N, 119.54245 W), and 3) 2.3 km outside the MPA boundary ("outside site"; 34.03837 N, 119.5253 W; Fig 1). At each of these three sites, we sampled directly along a 100 m fixed transect used by the Kelp Forest Monitoring Program for visual monitoring, using a GPS to ensure transects overlapped [4]. We collected three replicate 1 L water samples from three locations on each transect, totaling nine spatially structured replicates per site. Due to fieldwork logistical challenges, each site was sampled on a different day with a maximum of 72 hours between sampling events.

We collected seawater samples from 10 m below the surface and 1 m above the benthos using a 4 L Niskin bottle deployed from the UCLA RV Kodiak [24]. From each Niskin deployment, we transferred a single liter of seawater to an enteral feeding pouch and conducted gravity filtration through a sterile 0.22 μm Sterivex cartridge (MilliporeSigma, Burlington, MA, USA) in the field [25]. Additionally, we processed three field blanks as a negative control that consisted of 1 L of distilled water following the method above. Finally, we dried Sterivex filters using a 3 mL syringe and then capped and stored the filters at -20˚C for DNA laboratory work back at UCLA [26].

### DNA extraction and library preparation

We extracted eDNA from the Sterivex cartridge using the DNAeasy Tissue and Blood Kit (Qiagen Inc., Germantown, MD) following modifications of Spens et al. (2017). We PCR amplified the extracted eDNA using the MiFish Universal Teleost *12S* primer (Miya et al., 2015) with Nextera modifications following PCR and the library preparation methods of Curd et al. (2019) (See S1 Appendix for supplemental methods). All PCRs included a negative control where molecular grade water replaced the DNA extraction. For positive controls, we used DNA extractions of grass carp (*Ctenopharyngodon idella*, Cyprinidae*)* and Atlantic salmon (*Salmo salar*, Salmonidae), both non-native to California. Libraries were sequenced on a MiSeq PE 2x300bp at the Technology Center for Genomics & Bioinformatics (University of California- Los Angeles, CA, USA), using Reagent Kit V3 with 20% PhiX added to all sequencing runs.

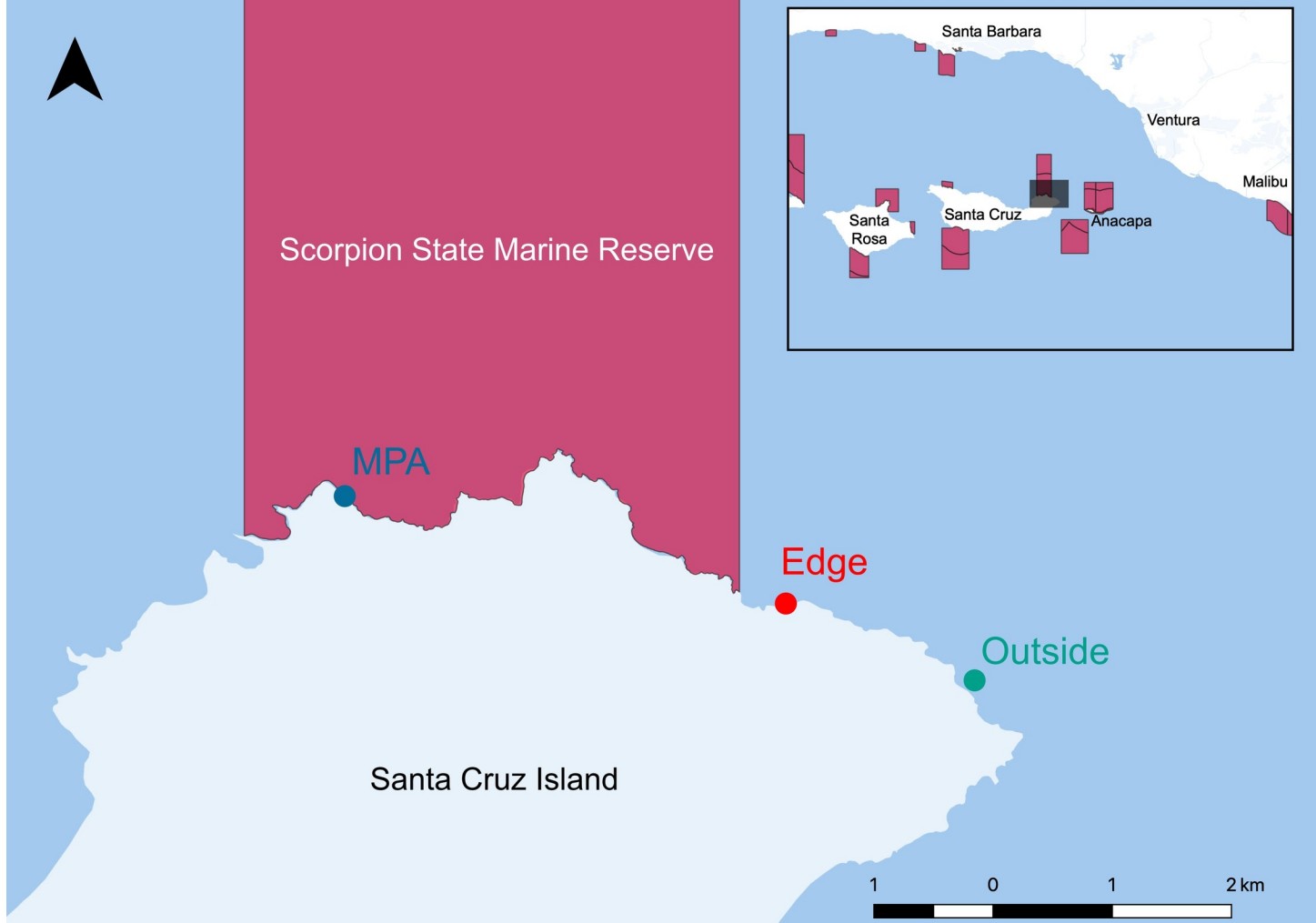

**Fig 1. Map of Scorpion State Marine Reserve off Santa Cruz Island, CA, USA.** The map was generated using the free and open source software QGIS version 3.0.

## Bioinformatics

To determine community composition, we used the *Anacapa Toolkit* (version: 1) to conduct quality control, amplicon sequence variant (ASV) parsing, and taxonomic assignment using user-generated custom reference databases [27]. The *Anacapa Toolkit* sequence QC and ASV parsing module relies on *cutadapt* (version: 1.16) [28], *FastX-toolkit* (version: 0.0.13) [29], and *DADA2* (version 1.6) [30] as dependencies and the *Anacapa classifier* modules relies on *Bowtie2* (version 2.3.5) [31] and a modified version of *BLCA* [32] as dependencies. We processed sequences using the default parameters and assigned taxonomy using two *CRUX*-generated reference databases. We first assigned taxonomy using the FishCARD California fish specific reference database [33]. Second, we used the *CRUX*-generated *12S* reference database supplemented with FishCARD reference sequences to assign taxonomy using all available *12S* reference barcodes to identify any non-fish taxa. We note that *CRUX* relies on *ecoPCR* (version: 1.0.1) [34], *blastn* (version: 2.6.0) [35], and *Entrez-qiime* (version: 2.0) [36] as dependencies.

Raw ASV community table was decontaminated following Kelly et al. (2018) and McKnight et al. (2019) (See S1 Appendix) [37]. We chose a site occupancy cutoff score of 84% which

corresponded with the minimum occupancy rate observed for three detections out of nine PCR replicates at a given location sampled. We then transformed all read counts into an eDNA index for beta-diversity statistics [15]. All non-fish species (mammals and birds) were removed prior to final analyses.

### eDNA data analysis

To test for alpha diversity differences, we compared total species richness for each site using an Analysis of Variance (ANOVA) and subsequent Levine's test for equality of variance [38].

To determine whether our eDNA sampling design was sufficient to fully capture fish community diversity, we created species rarefaction curves using the *iNext* package (version 2.0.2) [39]. We then compared species coverage estimates between each site, with and without site occupancy modeling, and using all three 1 L replicates taken at three locations along a 100 m transect (n = 9) as well as only three 1 L biological replicates (n = 3). We ran a piecewise regression analysis to identify breakpoints in the rate of species diversity found per sample collected using the *R* packaged *segmented* (version 1.3) [40].

To test for differences among fish communities, we calculated Bray-Curtis similarity distances on the eDNA index scores between all samples (See S2 Appendix for Supplemental Results) [22]. Specifically, we tested for the difference in community similarity variance between our three sites using an *adonis* PEMANOVA (*vegan* version: 2.4.2) [38], followed by a companion multivariate homogeneity of group dispersions test (BETADISPER) [38]. Both the PERMANOVA and BETADISPER were run using the following model: eDNA Index ~ Site + Location. We also visualized community beta diversity using non-metric multidimensional scaling (NMDS) [38]. To further investigate which species were driving eDNA community differences among sites, we conducted constrained analysis of principle components (CAP) [38].

### Visual underwater census methods

To assess fish communities using underwater visual census techniques, SCUBA divers from the Kelp Forest Monitoring Program followed standard survey protocols following Kushner et al. (2013). These protocols include survey types: visual fish transects, roving diver fish counts, and 1 $m^2$ quadrats. The visual fish transects targeted 13 indicator species of fish on visual fish transects recording the counts of adults and juveniles. This protocol consists of performing 2 m x 3 m x 50 m transects along the 100 m permanent transect. During roving diver fish count surveys all positively identified species are recorded. This protocol consists of 3–6 divers counting all fish species observed during a 30 minute time period, covering as much of the 2000 $m^2$ of bottom and entire water column as possible. The 1 $m^2$ quadrat records three small demersal species of fish. All visual surveys occurred along a permanent 100 m transect at each site and were conducted within two weeks of eDNA sampling (See S1 Appendix).

### Comparison of eDNA and visual underwater census methods

We compared species detected by eDNA and underwater visual census approaches across corresponding transects at each site. We identified core taxa that were shared across all sites for eDNA and visual survey methods. In addition, we identified species that eDNA methods failed to detect but were observed in visual census surveys and vice versa. Given the few numbers of sites (n = 3), we were unable to robustly compare abundance estimates between methods.

## Results

### eDNA results

We generated over four million reads that passed quality control. The *Anacapa Toolkit* identified 2,906 ASVs from 3,091,063 reads representing 27 samples and eight controls. After the second decontamination step, however, totals reduced to 441 ASVs and 2.23 million reads (S1-S3 Tables in S1 File).

Combined, eDNA metabarcoding successfully detected 42 fish taxa, representing 40 unique species, 39 genera, 28 families, and two classes (S1-S3 Tables in S1 File). eDNA detected 31 species within the MPA, 36 at the edge, and 38 species outside the MPA. The three sites shared a core group of 29 taxa including bony fish and one species of ray (Fig 2) (S4 Table in S1 File). Of these taxa, 18 species are associated with rocky reef habitat, five species are associated with sandy bottom habitat, four species are pelagic-neritic, and two species are pelagic-oceanic.

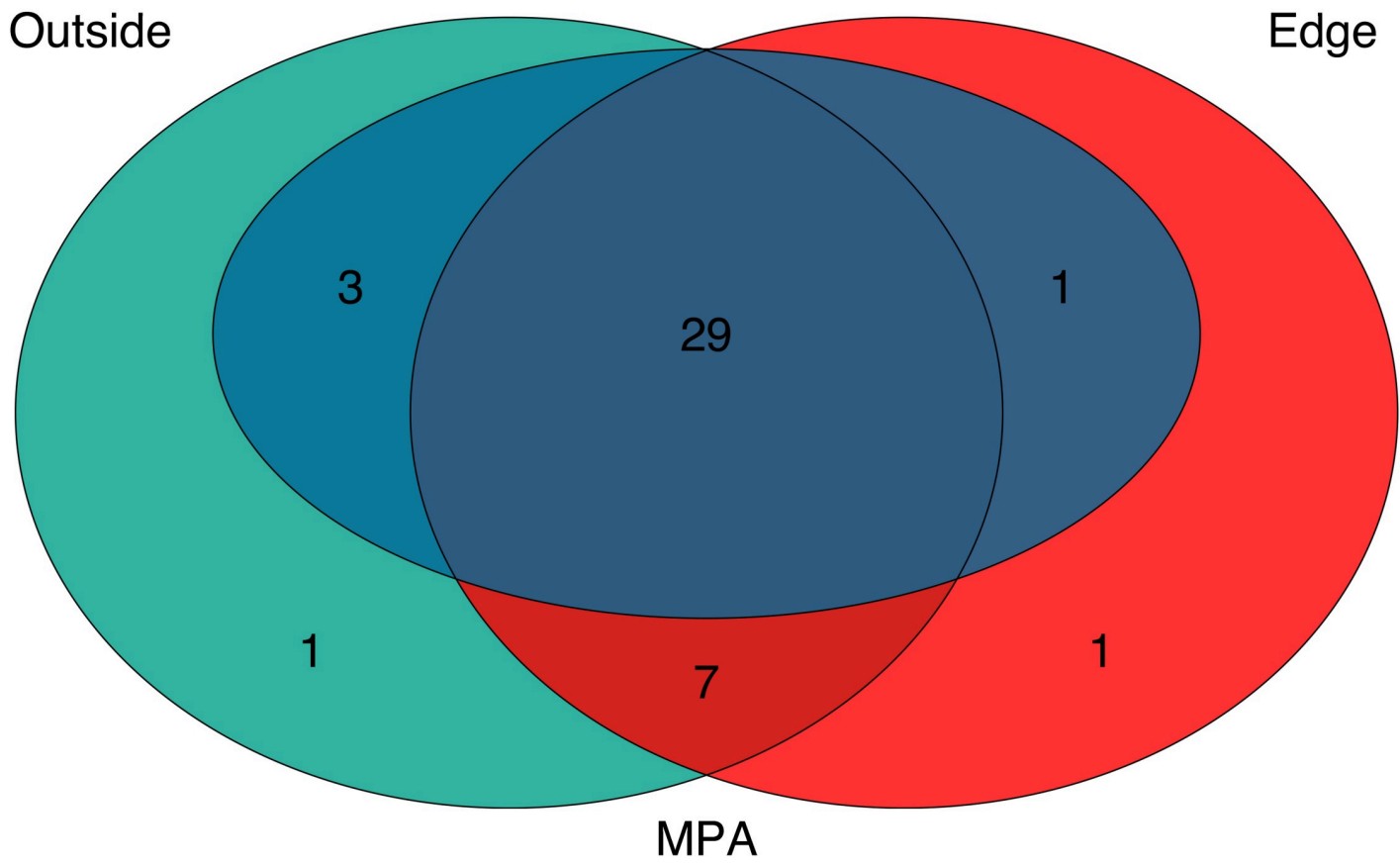

**Fig 2. Venn diagram of fish species detected with eDNA.**

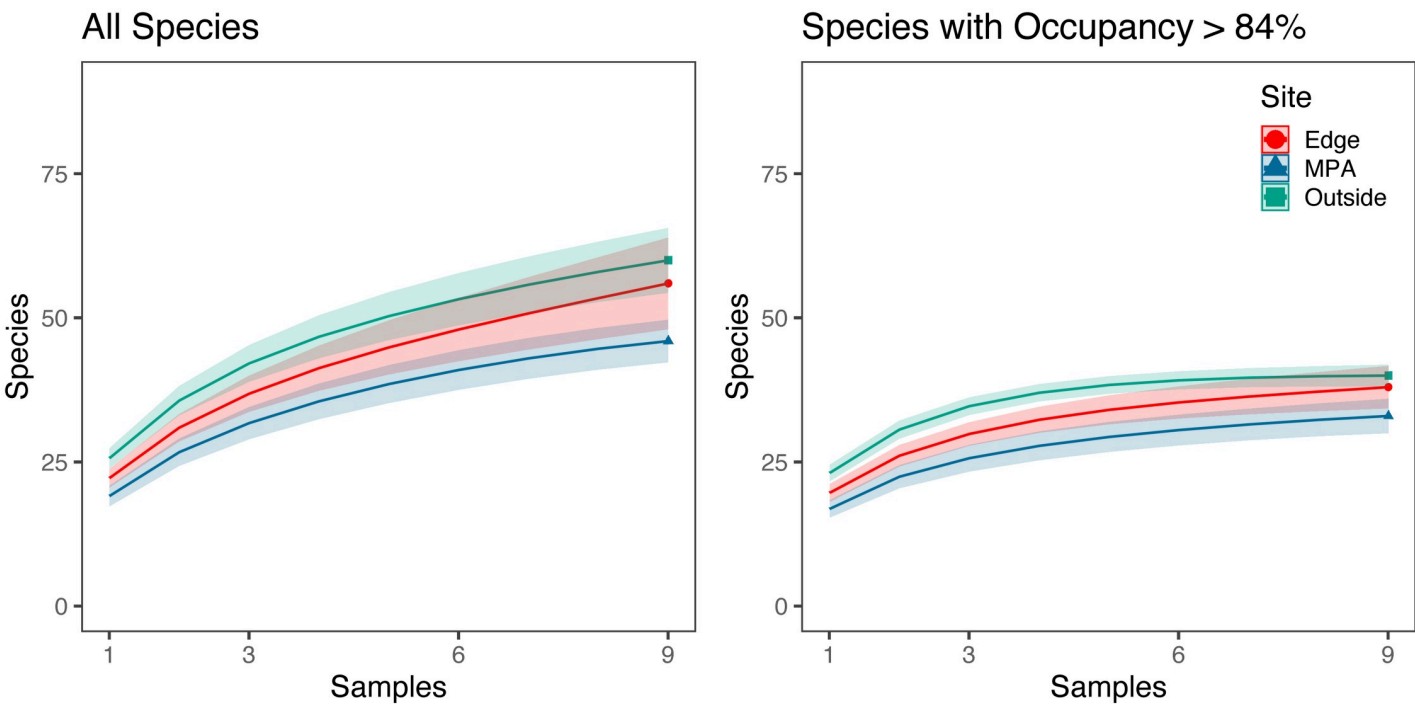

**Fig 3. Species rarefaction curves. a)** Species rarefaction curves for all fishes found at each site across three 1 L replicates taken at three locations along a 100m transect. **b)** Species rarefaction curves for fish species with occupancy rates above 84% found at each site across three 1 L replicates taken at three locations along a 100 m transect. Sample coverage estimates were higher for species with occupancy rates above 84% (96.7–99.8%) than for all species (88.9%-94.4%). For species with occupancy rates above 84% sample coverage estimates ranged from 87.3–90.0% for only three 1 L replicates.

Species rarefaction curves showed that sampling at each site (n = 9) was insufficient to capture all species diversity (Fig 3). Sample coverage estimates from eDNA results before filtering by site occupancy modeling filters were 94.4%, 88.9%, and 93.0% for the MPA, edge, and outside sites, respectively. Coverage estimates dropped to 80.2%, 80.0%, 82.0% for the MPA, edge, and outside sites, respectively, when only three 1 L samples per site were used. Piecewise regression analysis showed a transition from exponential to linear increase in species detected per replicate between three and four replicate water samples per site (3.35–3.47) with subsequent diminishing sample coverage returns with the addition of more samples. In contrast, species diversity was near saturated (96.7%, 96.4%, and 99.8% for the MPA, edge, and outside sites, respectively) when applying a site occupancy rate above 84% and using three 1 L replicates taken at three locations along a 100 m transect. However, using only three samples, sample coverage dropped to 87.1%, 90.3%, 88.9% for the MPA, edge, and outside sites, respectively.

Analyses showed a significant difference in the total number of observed species across sites, with the site outside the MPA having significantly higher richness than both the edge and MPA sites (ANOVA, p<0.001, Levine's test p> 0.5). Observed species differences between sites were partially driven by the presence of non-rocky reef taxa (48.4%, 5/13), primarily pelagic, mobile, sandy bottom, and intertidal species. Moreover, there were also significant differences in fish communities among the three sites as well as among the three sampling locations along each of the three transects (PERMANOVA p<0.001, betadisper p>0.05). Location along the transect explained 27.5% of the total variance while site (e.g. inside, edge and outside MPA) explained 22.5% of the total variance; 50.0% of the total variance was unexplained.

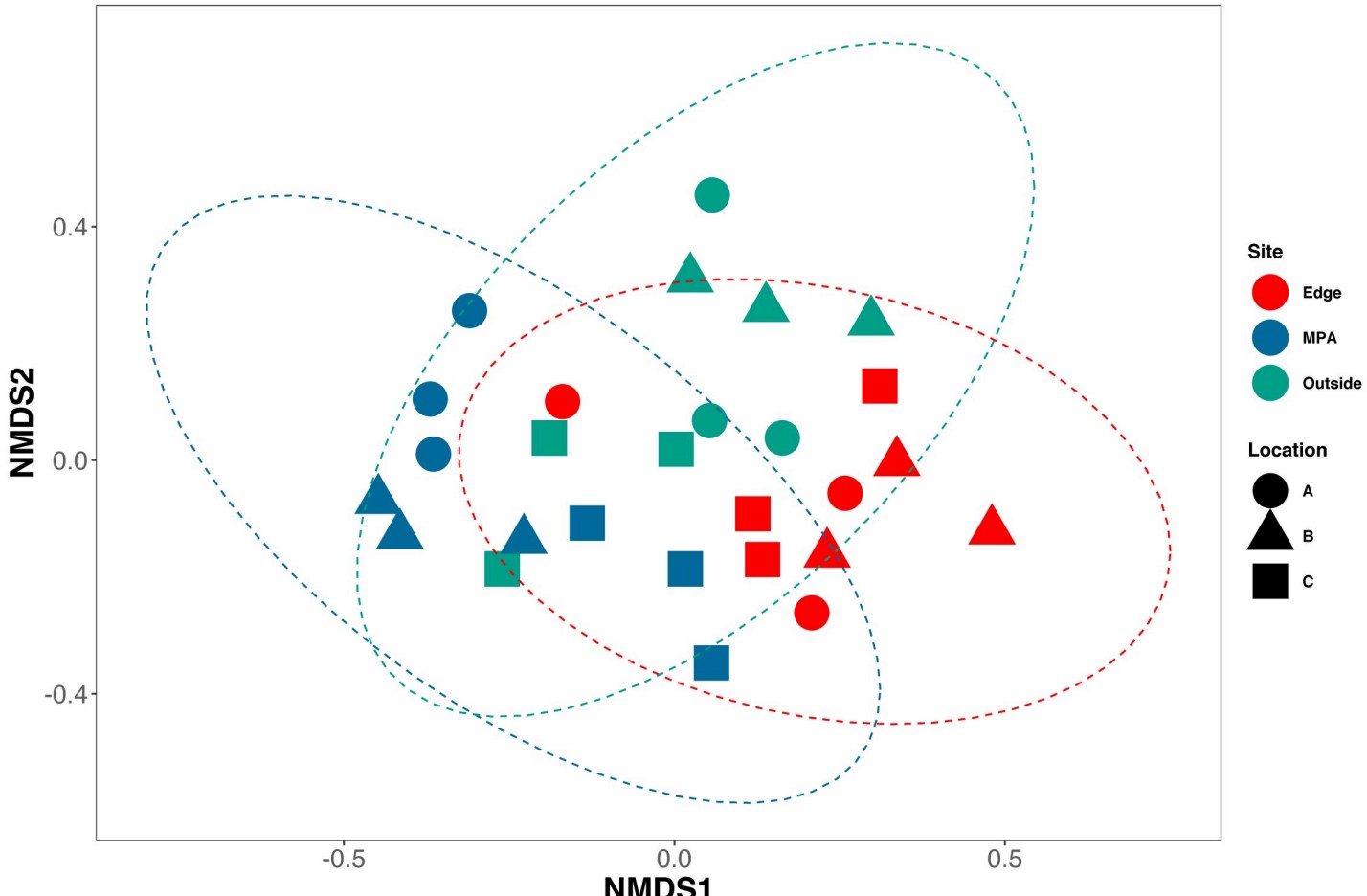

**Fig 4. NMDS of Bray-Curtis dissimilarities.** Bray-Curtis dissimilarities were calculated between all samples using only species with occupancy rates over 84%. Samples from Sites (colors) and locations (shapes) are similar to each other (NMDS, Stress = 0.21).

NMDS ordination showed weak clustering of samples by both location and site (NMDS, Stress 0.21; Fig 4). Constrained analysis of principle components (CAP) found significant differences in species assemblages between samples collected at different sites and locations (CAP, p<0.001) (Fig 5), further indicating difference in eDNA signatures across sites and locations. CAP analysis identified seven taxa with the strongest differences between sites. The MPA site had higher eDNA index scores of opaleye (*Girella nigricans*, Kyphosidae*)* and kelp bass (*Paralabrax clathratus*, Serranidae). The edge site had higher index scores of blacksmith (*Chromis punctipinnis*, Pomacentridae) and fantail flounder (*Xystreurys liolepis*, Paralychthyidae). The site outside the MPA had higher index scores of giant black sea bass (*Stereolepis gigas*, Polyprionidae), Pacific barracuda (*Sphyraena argentea*, Sphyraenidae), and topsmelt (*Atherinops affinis*, Atherinopsidae). Results using Jaccard-binary dissimilarities were highly concordant (S2 Fig, See S2 Appendix).

## Visual census surveys results

Across all three sites, 25 bony fish species were recorded using underwater visual censuses, representing 20 genera, 13 families, and one class (Fig 6) (S5 Table in S1 File), 11 of which

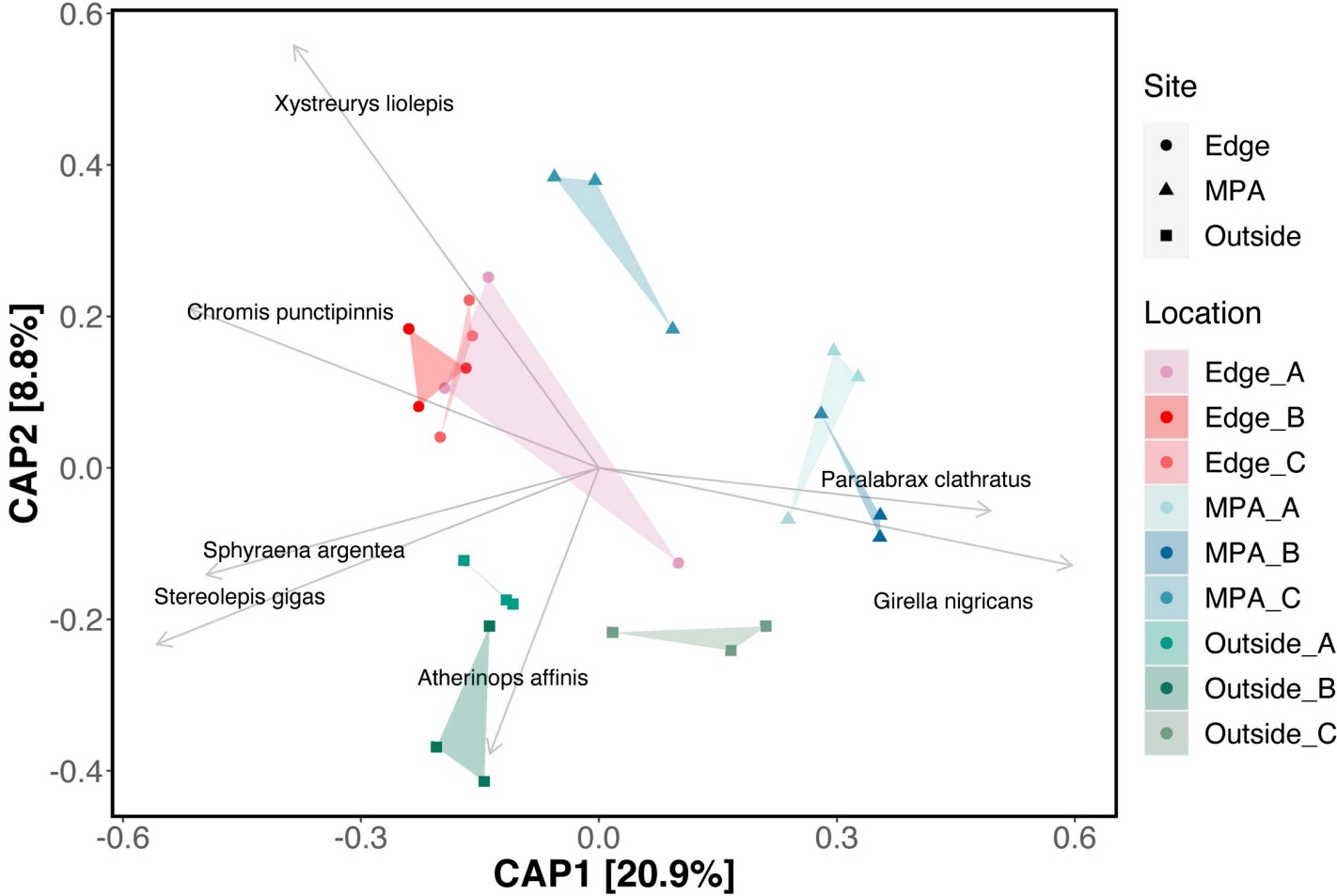

**Fig 5. Constrained analysis of principle components (CAP) ordination.** Bray-Curtis dissimilarities were calculated between all samples using only species with occupancy rates over 84%. Samples from sites and locations within sites were used as independent variables. Site and locations within sites are significantly more similar to each other (CAP, p<0.001). Sites (shapes) and Locations (colors) are plotted against CAP1 and CAP2 axes. Arrows correspond to direction and strength (length) of each species. Only the top seven species with CAP distances greater than 0.40 were plotted.

were shared across all three sites (S6 Table in S1 File). Within the MPA site, visual census methods detected 21 unique species, 18 genera, and 11 families. At the edge site, visual census methods detected 18 species, 16 genera, 11 families, and four classes. Lastly, at the outside site visual census methods detected 13 species, 13 genera, 10 families, and four classes. Of all taxa observed in visual census methods, 24 species were associated with rocky reef habitat and one species was pelagic-neritic. The pelagic-neritic species, top smelt (*Atherinops affinis*, Atherinopsidae), was only found in the MPA site.

On average, roving diver fish counts recorded 17.6 species per replicate survey (Range: 10–22). Visual fish counts recorded an average 7.8 species per replicate survey out of the 13 indicator species (Range: 5–10). 1 m$^2$ quadrats recorded an average 2.3 species of 3 target species (Range: 1–3).

## Comparison of eDNA and visual census surveys

eDNA detected 76% (19 out of 25) of species observed during all combined National Park Service transect surveys (S5-S6 Table in S1 File). eDNA failed to resolve *Lythrypnus dalli*

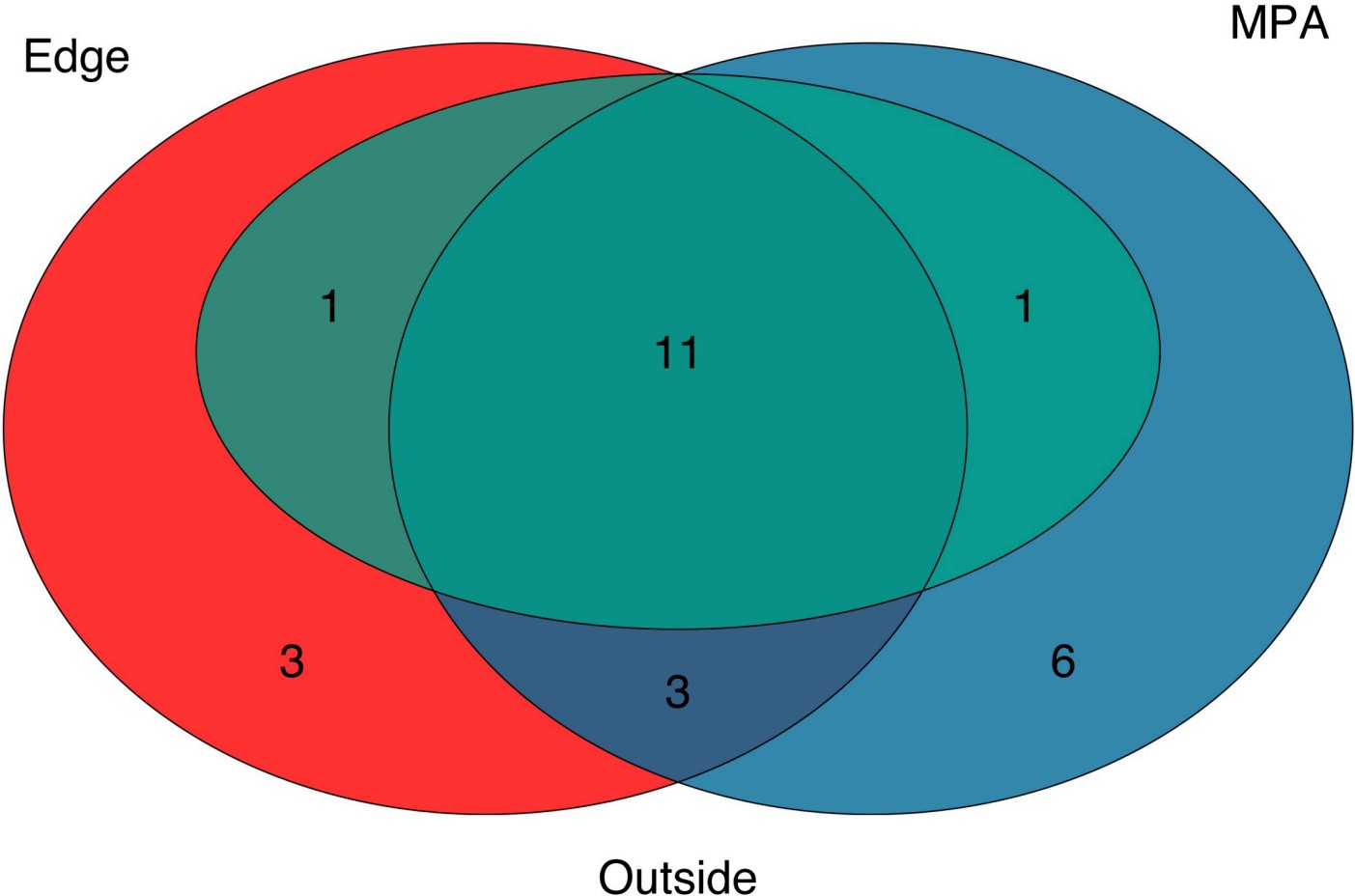

**Fig 6. Venn diagram of species observed from visual SCUBA surveys.**

(Gobiidae), *L. zebra*, *Sebastes atrovirens* (Sebastidae), *S. auriculatus*, *S. chrysomelas*, and *S. serranoides* to species level. At the genus level, eDNA performed markedly better recovering 95% (19 out of 20) of genera observed during under water censuses. The remaining genus *Lythrypnus* was detected prior to site occupancy modeling, but occurred in only one replicate at two separate sites. In addition to the above, eDNA recovered 23 species that were not recorded during the visual censuses conducted by the National Park Service. Of these, all were California fish species previously recorded in Kelp Forest Monitoring Program surveys (S7 Table in S1 File), but not observed during our paired surveys.

There were few conspicuous differences in species observed across sites, with visual census results identifying 11 common taxa across all sites (S6 Table in S1 File). Of these, 10 were also found to be common across all sites using eDNA methods; the remaining species, *Lythrypnus dalli*, was detected by eDNA but was removed following site occupancy modeling. Species

richness from visual census data showed that fish diversity was highest within MPA (n = 21), lowest outside the MPA (n = 13) and intermediate (n = 18) on the edge of the MPA. This is in contrast to the eDNA results which found the opposite pattern with higher species richness outside (n = 38) and on the edge (n = 36) of the MPA and lower species richness inside the MPA (n = 31).

## Discussion

Results demonstrate the power of eDNA for detecting a broad range of fish biodiversity in California kelp forest ecosystems, providing more detailed species inventories needed for marine ecosystem monitoring [9]. eDNA detected significant differences in fish communities inside, on the edge of, and outside of the Scorpion State Marine Reserve, even though the closest sites were no more than 500 m apart. Even within each of these sampling sites, eDNA distinguished among sample locations separated by only 50 m, highlighting the sensitivity of eDNA in capturing local fish communities, and matching previous studies showing fine-scale spatial resolution of eDNA signatures [12].

Importantly, eDNA captured 76% of fish diversity observed during visual surveys, despite species rarefaction indicating insufficient sampling. In total, eDNA only failed to identify six of 25 fish species observed during visual surveys, four of these being rockfish (*Sebastes*, Sebastidae), a taxon that *12S* barcoding cannot distinguish to species [33]. This small deficiency was offset by detecting an additional 23 fish taxa not recorded during paired Kelp Forest Monitoring Program visual monitoring, representing an important advantage of eDNA. Because sampling can be obtained easily and processed economically, eDNA could allow for more frequent monitoring, expanding the scope of MPA monitoring programs while providing greater personnel safety.

### The utility of eDNA for MPA monitoring

Despite the limited eDNA sampling design and the inability the *12S* barcode to distinguish species of rockfish and gobies, eDNA largely recovered the same taxa observed in visual census surveys. This strong concordance likely stems from high eDNA detection probabilities lasting only a few hours [20], such that eDNA captures marine communities that were recently present [21]. The similarity of eDNA and visual surveys is even more remarkable given that eDNA and visual surveys were taken two weeks apart, a result that strongly suggests that fish diversity captured by eDNA is truly representative of fish communities and their differences inside and outside the Scorpion State Marine Reserve [22].

In addition, eDNA recorded an additional 23 species not recorded from visual surveys, but have been previously reported in other Kelp Forest Monitoring Program surveys (S7 Table in S1 File). Importantly, these taxa included species of significant management concern such as the critically endangered (IUCN) giant black seabass (*Stereolepis gigas*, Polyprionidae) and important commercial species like yellowtail amberjack (*Seriola lalandi*, Carangidae). Additionally, although we focused on teleost fishes, our eDNA data also included elasmobranchs, marine mammals, and marine birds, taxa that play important roles in nearshore rocky reef ecosystems, but that can be difficult to survey and monitor [9]. The expanded taxonomic coverage and the ability to detect rare, or cryptic taxa is a significant advantage of eDNA over traditional visual surveys, expanding the scope of MPA monitoring by capturing entire communities rather than a selected subset of taxa.

Key to MPA monitoring is the ability to distinguish among communities inside and outside of the MPA. Not only did eDNA detect significant differences inside and outside the MPA, it could also differentiate among samples taken 50 m apart. This result adds to a growing

literature that shows the fate and transport of eDNA in marine environments is relatively lim-ited in space and time [20,41,42], and highlights the suitability of eDNA for comparing inside and outside of even relatively small MPAs [12].

While eDNA found significant differences inside and outside of the MPA and provided data on more taxa than visual survey methods, a key question remains as whether eDNA pro-vides equivalent biomonitoring data for lower cost and effort [43,44]. We note that estimating the exact costs of visual surveys and eDNA surveys is challenging, given the stark differences in equipment, training, and infrastructure required to support these different biomonitoring efforts [45]. Although in this case we found eDNA was less expensive (S8-S9 Tables in S1 File), we note this may not always be the case depending on the locations and species surveyed as well as the expertise of the individuals involved [9]. Regardless, what is important is that eDNA allows individuals with no dive experience or taxonomic training to obtain comparable, and at times, more complete biomonitoring results than from conducting SCUBA-based visual surveys [8,46–48]. Furthermore, this opens up sampling to times and locations where visual surveys cannot be conducted [9].

In addition to above, eDNA has other significant advantages. It can potentially detect inva-sive species, even when rare [8]. Sequence data from eDNA provides an annual snapshot of standing genetic diversity, providing the ability to monitor changes over time [9]. Similarly, in species with population structure, eDNA could provide evidence of range shifts associated with climate change [49]. Importantly, given eDNA metabarcoding samples can be preserved and archived, eDNA samples can be reanalyzed in the future with improved metabarcoding methods to answer additional hypotheses and environmental monitoring goals [8]. Combined, the above advantages of eDNA suggest that even if eDNA metabarcoding is not viewed as a full replacement for visual surveys, the power of this method, and it's ease of sampling and affordability argue for using eDNA as a critically important complementary tool to greatly expand current monitoring activities.

## Limitations and caveats of eDNA

Although this and other studies highlight the promise of eDNA for monitoring marine ecosys-tems, there are also important limitations. One key limitation is the lack of universal barcode loci. Four of the six undetected species in this study were rockfish in the genus *Sebastes*. While the MiFish *12S* metabarcoding primers have broad utility in vertebrates, rockfishes are a recent adap-tive radiation [50] with a highly conserved 12S sequence, resulting in the inability to distinguish among rockfish ASVs. Identifying rockfish to species using eDNA approaches is critical for MPA monitoring efforts in California as *Sebastes* are important for commercial and recreational fisher-ies [51] and play a wide array of functional and ecological roles in nearshore ecosystems [50].

In addition, eDNA failed to detect two gobies, *Lythrypnus dalli* and *L. zebra* (Gobiidae). Previous efforts to barcode *L. dalli* for the FishCARD reference database found two insertions not found in any other California goby, including the sister species *L. zebra* [33]. Thus, primer mismatch may have limited the amplification and detection of some *L. dalli* in our eDNA sam-ples. Alternatively, the eDNA methods employed here may not be suited for small, crevice-dwelling fish species such as gobies. Species of *Lythrypnus* rarely leave the reef boundary layer [52]. As such their eDNA maybe entrained close to the reef, resulting in hyper-spatial variabil-ity of eDNA signatures [12]. More work is necessary to determine whether eDNA can reliably detect species living in interstitial reef habitat. This limitation, however, is not unique to eDNA as the Kelp Forest Monitoring Program employs 1 m$^2$ quadrat surveys, specifically designed to capture these taxa. Likewise, eDNA surveys that specifically sample within the boundary layer may be needed to survey benthic cryptic species.

Another limitation of eDNA is standardizing processing techniques, including the spatial design of field sampling, number of replicates, and sequencing depth [8,15,19]. The three replicate water samples taken from a single location and time recovered only 88.3% of the species present based on modeled species coverage estimates of species with at least 84% occupancy. This value increased to near saturation (>97.6%) by sampling three replicate water samples from three locations along a 100 m transect. That said, rarefaction curves indicated that additional sampling would have recovered more species. These results provide important benchmarks for replication and sampling efficiency within nearshore marine environments and highlight the need to adjust sampling intensity and replicates, depending on the questions to be addressed with eDNA.

Despite not achieving saturation with our sampling design, we did observe a transition from exponential to linear addition of species detections with additional sampling similar to that previously demonstrated in mesocosm experiments [19]. This shift likely reflects the biological reality of eDNA within marine ecosystems, with a few taxa being abundant and a long tail of low abundant species [15]. As such, while only a few replicates are needed to capture local core species diversity, high technical (PCR) and biological (bottle) replication may be required to saturate species detection [19]. Thus, if the goal is to detect rare species, it is imperative to increase sample number, an unsurprising result given the reality of detection probabilities of rare taxa [18]. Despite this caveat and our relatively limited number of sample replicates, we still detected rare species such as giant black seabass (*Stereolepis gigas*, Polyprionidae) suggesting that eDNA is likely still superior to visual techniques at rare species detection [10].

## Importance of site occupancy modelling

Site occupancy modeling showed that almost all species with occupancy rates higher than 84% were common Southern California kelp forest species [53]. In contrast, almost all pelagic and intertidal species that should not be present in a kelp forest had low occupancy rates and were detected only in a single bottle replicate (S1-S2 Tables in S1 File). These low occupancy detections cannot be contamination because they did not occur in field or laboratory controls; instead, they likely represent eDNA transported between habitats [16]. Regardless, site occupancy modeling removed the vast majority of unexpected kelp forest fishes, highlighting its value for determining true species detections in a rigorous and repeatable way [18,19], aiding in the interpretation and comparison of eDNA results.

While site occupancy modelling removed non-kelp forest taxa (e.g. blue whale; *Balaenoptera musculus* (Cetacea); California sea lion, *Zalophus californianus* (Otariidae); pelagic cormorant *Urile pelagicus* (Phalacrocoracidae); S10 Table in S1 File), it also removed some kelp forest species (e.g. zebra goby, *L. dalli*, Gobiidae; swell shark, *Cephaloscyllium ventriosum*, Scyliorhinidae; zebra-perch *Hermosilla azurea*, Embiotocidae; California angel shark, *Squatina californica*, Squatinidae). These results highlight the need for increased replication depending on the management question, just as it may require more visual surveys to observe numerically rarer taxa, such as sharks. Although the ability of eDNA to detect marine mammals and birds is useful, visual observations maybe more effective depending on the taxa, suggesting that complementary methods may yield the most effective sampling regime [54].

## Diversity inside and outside MPAs

Traditional visual surveys most often report higher biodiversity and biomass inside MPAs [55], including Scorpion State Marine Reserve [6]. However, our results surprisingly indicate lower diversity inside the MPA. This paradoxical result is partially explained by the inability of

eDNA to resolve *Sebastes* species that were visually observed inside (n = 3) and on the edge of the MPA (n = 1), but not outside. However, *Sebastes* only accounts for some of the differences inside and outside of the MPA, suggesting that other explanations are required.

One potential source of error could be sampling design of visual and eDNA methods. Time limited SCUBA surveys may not capture species richness as well as eDNA surveys outside the MPA where fish abundance is lower, reducing detection probabilities. Similarly, the nine eDNA samples taken in each region may not capture true patterns of species richness (but see S1 Fig). Distinguishing among these possibilities may be possible by using visual survey protocols that increase transect replication and taxonomic focus [56,57] and using eDNA protocols with increased spatial sampling replication to mitigate patchiness of eDNA dispersion across a reef [8] or by increasing technical PCR replication to reach full species saturation [19].

While the above highlights the difficulty of comparing two surveying methods with imperfect detection rates [54,58], there may be a simpler explanation for eDNA capturing more diversity outside the MPA. Inside an MPA where resident kelp forest fishes are very abundant, the concentration of local fish eDNA is likely very high. In contrast, for transient species passing through these habitats, or for eDNA advected from adjacent habitats, eDNA concentrations would likely be relatively low. However, outside the MPA, where local kelp forest fish are less abundant, both local and transient/advected eDNA concentrations would be low. Because PCR is a probabilistic process, the strongly skewed concentrations of local kelp forest taxa eDNA inside the MPA may reduce the probability of amplifying and detecting rare taxa, local or transient. In contrast, outside the MPA where all eDNA signals are low, the probability of detecting transient and/or advected eDNA would increase.

Support from this hypothesis comes from examining taxa recovered outside the MPA that were not detected inside the MPA. In total, 38.5% of taxa detected outside the MPA were non-rocky reef species such as Yellowtail amberjack (*Seriola lalandi*, Carangidae), California clingfish (*Gobiesox rhessodon*, Gobiesocidae), and Fantail flounder (*Xystreurys liolepis*, Paralichthyidae). These species occasionally pass through rocky reef environments, but their eDNA could also be transported from nearby pelagic, intertidal, and sandy bottom ecosystems [16,42]. In either case, eDNA concentrations would be relatively low, with low detection probabilities inside the MPA but comparatively higher detection probabilities outside. In addition, the MPA site had high kelp abundance while kelp was largely absent outside of MPA [59]. Because kelp creates a three dimensional structure that dampens cross reef flow [60], advection of foreign eDNA should be more likely outside the MPA, potentially increasing the probability of detecting non-local eDNA signatures.

Whatever the cause, the paradoxical pattern of species richness observed in this study highlights that eDNA data must be interpreted judiciously [61]. Metabarcoding methods often perform unexpectedly when DNA concentrations are low, increasing the probability of sequencing rare species [8]. Thus additional ecological metrics to species richness, ones that are more representative of ecological patterns and processes, are needed to optimally interpret eDNA results [15]. These results ultimately highlight the value of ground truthing eDNA results with visual surveys in novel applications to ensure proper interpretation of results [54]. However, they also highlight that comparing survey methods with imperfect detection efforts in the field is challenging as there is no "gold standard" for ecosystem biomonitoring [54,58].

## Conclusion

Marine protected areas are indispensable tools for protecting marine ecosystems and effective monitoring is paramount to their success [1]. Our results demonstrate that eDNA can distinguish fish assemblages inside and outside MPAs, and can detect other vertebrates, like marine mammals and birds, of special conservation concern.

Given its power, ease of sampling and relative affordability, eDNA could provide critical added benefits of repeated temporal or expanded spatial sampling of marine protected areas. In particular, eDNA metabarcoding can overcome many of the current limitations of visual monitoring, increasing sampling frequency and expanding monitoring beyond a small subset of "important" focal taxa. Such expanded monitoring would improve our ability to understand the ecological processes, human impacts, and management strategies that affect marine communities that MPAs are designed to protect.

However, important aspects of eDNA remain unresolved, particularly with respect to estimating abundance and biomass in marine ecosystems [62]. Because eDNA cannot currently provide these critical density measurements needed for effective fisheries management [63,64], eDNA should not be viewed yet as a wholesale replacement for visual monitoring, but instead as a powerful complementary tool. There will always be value in the direct observation by divers in particular for biomass estimates, informing size class distributions and sex ratios [23]; however, eDNA provides an important way to make visual surveys more comprehensive and efficient. By replacing aspects of underwater visual surveys, eDNA could reduce the dive time per site, allowing more sites to be surveyed more frequently or improve overall biodiversity estimates. Additionally, field collection of eDNA could be completed in a week, allowing for surveys to occur during short periods of good weather in the winter when full visual surveys would be impossible. As such, eDNA could greatly expand current monitoring activities across space, time, and depth, providing resource managers critical information on the response of MPAs to changing environments and management practices.

## Supporting information

**S1 Appendix. Supplemental methods.**
(DOCX)

**S2 Appendix. Supplemental results.**
(DOCX)

**S1 Fig. Species richness sequence depth rarefaction.**
(TIF)

**S2 Fig. NMDS of Jaccard-binary dissimilarities.** Jaccard-binary dissimilarities were calculated between all samples using only species with occupancy rates over 84%. Samples from Sites (colors) and locations (shapes) are similar to each other (NMDS, Stress = 0.23).
(TIF)

**S1 File.**
(XLSX)

## Acknowledgments

We thank the Channel Islands National Park Service and Kelp Forest Monitoring Program divers for conducting underwater visual censuses. We gratefully acknowledge Tiara Moore, Nick Schooler, Rachel Turba, and Lucia Bertero for their tireless help in the field collection of eDNA samples. We also acknowledge Taylor Ely and Onny Marwayana for assistance with laboratory work.

## Author Contributions

**Conceptualization:** Zachary Gold, Joshua Sprague, David J. Kushner, Paul H. Barber.

**Data curation:** Zachary Gold, Joshua Sprague.

**Formal analysis:** Zachary Gold, Joshua Sprague, David J. Kushner, Erick Zerecero Marin.

**Funding acquisition:** Zachary Gold, Paul H. Barber.

**Investigation:** Zachary Gold, Joshua Sprague, David J. Kushner, Erick Zerecero Marin.

**Methodology:** Zachary Gold, Erick Zerecero Marin, Paul H. Barber.

**Project administration:** Zachary Gold, Joshua Sprague, David J. Kushner, Erick Zerecero Marin, Paul H. Barber.

**Resources:** Zachary Gold, David J. Kushner, Paul H. Barber.

**Software:** Zachary Gold.

**Supervision:** Zachary Gold, David J. Kushner, Paul H. Barber.

**Validation:** Zachary Gold.

**Visualization:** Zachary Gold.

**Writing – original draft:** Zachary Gold.

**Writing – review & editing:** Zachary Gold, Joshua Sprague, David J. Kushner, Erick Zerecero Marin, Paul H. Barber.

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
