## [Decision Letter · Decision Letter 0]

5 Nov 2020

PONE-D-20-25777

eDNA metabarcoding as a biomonitoring tool for marine protected areas

PLOS ONE

Dear Dr. Gold,

Thank you for submitting your manuscript to PLOS ONE. After careful consideration, we feel that it has merit but does not fully meet PLOS ONE’s publication criteria as it currently stands. Therefore, we invite you to submit a revised version of the manuscript that addresses the points raised during the review process.

I tend to agree with the comments/suggestions made by reviewer #1. In your revision please specifically address your interpretation/explanation of the species richness based on your comparative estimates using SCUBA survey and eDNA, and consider your conclusion in a broader perspective.

We look forward to receiving your revised manuscript.

Kind regards,

Andrea Belgrano, Ph.D.

Academic Editor

PLOS ONE

2. In your Methods section, please provide additional location information of the sampling sites, including geographic coordinates for the data set if available.

3. We note that you are reporting an analysis of a microarray, next-generation sequencing, or deep sequencing data set. PLOS requires that authors comply with field-specific standards for preparation, recording, and deposition of data in repositories appropriate to their field. Please upload these data to a stable, public repository (such as ArrayExpress, Gene Expression Omnibus (GEO), DNA Data Bank of Japan (DDBJ), NCBI GenBank, NCBI Sequence Read Archive, or EMBL Nucleotide Sequence Database (ENA)). In your revised cover letter, please provide the relevant accession numbers that may be used to access these data. For a full list of recommended repositories, see http://journals.plos.org/plosone/s/data-availability#loc-omics or http://journals.plos.org/plosone/s/data-availability#loc-sequencing.

4. We note that Figure 1 in your submission contain map images which may be copyrighted. All PLOS content is published under the Creative Commons Attribution License (CC BY 4.0), which means that the manuscript, images, and Supporting Information files will be freely available online, and any third party is permitted to access, download, copy, distribute, and use these materials in any way, even commercially, with proper attribution. For these reasons, we cannot publish previously copyrighted maps or satellite images created using proprietary data, such as Google software (Google Maps, Street View, and Earth). For more information, see our copyright guidelines: http://journals.plos.org/plosone/s/licenses-and-copyright.

(1) You may seek permission from the original copyright holder of Figure 1 to publish the content specifically under the CC BY 4.0 license.

Reviewers' comments:

Reviewer's Responses to Questions

**Comments to the Author**

1. Is the manuscript technically sound, and do the data support the conclusions?

Reviewer #1: Partly

2. Has the statistical analysis been performed appropriately and rigorously? 

Reviewer #1: Yes

3. Have the authors made all data underlying the findings in their manuscript fully available?

Reviewer #1: Yes

4. Is the manuscript presented in an intelligible fashion and written in standard English?

Reviewer #1: Yes

5. Review Comments to the Author

Reviewer #1: Gold and co-authors present a study comparing eDNA metabarcoding and SCUBA surveys to monitor fish populations in a marine protected area (MPA). The manuscript is well written, technically sound and adds to our knowledge comparing SCUBA and eDNA metabarcoding for applied monitoring of MPAs. The authors show similar species detected by both methods and demonstrate the eDNA metabarcoding can resolve geographically separated sites. However, the authors conclude that the greater estimated species richness outside of the MPA (contrary to expectations and SCUBA observations) detected by eDNA metabarcoding can be attributed to a swamping effect of MPA inhabiting fish. While this might well be true the authors present no data to support the claim. If the authors can provide more support for this argument, or provide a more broad explanation for the observations, then I would support publication of the manuscript. I provide an expansion of my major objection and some minor line specific comments below.

The observation of greater species richness in sites outside of the MPA has three possible explanations in my view. First, there might actually be greater species richness outside of the MPA and eDNA provides evidence for biases in the SCUBA surveys. Second, the sampling design is picking up highly localised species richness differences and including more than one site per site type may reveal the expected pattern. Or third, some methodological bias due to eDNA transport or in the workflow has resulted in an incorrect estimate of species richness. In lines 444-470 the authors seem to think the third option is the most likely, particularly focussing on the effect of eDNA transport in their discussion. I have detailed in a line specific comment below my objection to read depth of samples being listed as a possible explanation. This withstanding I think more broad overview of the possible explanations is required. We simply don’t have enough data to attribute the observations of greater species richness to loss of species detections in the MPA due to a swamping of the PCR with kelp taxa DNA.

Line 63-68: These observations are great but can we generalise these issues to all SCUBA-based monitoring programs?

Line 72: Perhaps “…largely limited to…” rather than ““…largely limited o…”

Line 155: I think the Anacapa Toolkit is a nice tool but it relies on loads of software packages that are not cited here. The authors should cite these packages to provide proper attribution to their authors, and also describe the version number of the toolkit and associated underlying software for reproducibility.

Line 159: Should the reference here be a number? If so I think it is number 29?

Line 167: In your data availability statement you mention that the code will be available on GitHub. Please upload this data before submitting the paper – it’s really valuable to be able to scan through code to follow a manuscript. I have received some excellent suggestions from reviewers on my own code and it shows you aren’t afraid for someone to look under the hood of your analysis!

Line 169: In my mind a Levine’s test examines equality of variance between groups? I think “homogeneity of dispersions” is also technically correct but this term applies more frequently when testing for between multivariate groups after/before a PERMANOVA procedure.

Line 180: This function is from vegan. Please cite vegan here or make it clear, please detail the version number of vegan and R used.

Line 261: What does this diagram look like when you use a binary index of species incidence like Jaccard? Does the eDNA index used to generate Bray-Curtis dissimilarity add lots of information to the separation of these samples?

Lines 449-452: This section is confusing to a reader. Does read depth matter here if species richness curves have saturated? If the authors believe they do then this is a problem that should be discussed further and potentially there should be some analysis that looks at the number of detections per read (or rarefied richness estimates explored). Personally I think the authors provide good evidence that they have sufficient read depth and adding this methodological caveat makes a reader begin to distrust the data unnecessarily.

Line 514: Please also upload the raw sequence data to a public repository (eg NCBI or ENA)

Supplementary Methods: Over eight PCR reactions per sample are required to saturate OTU detection (Doi et al. 2019, doi.org/10.1038/s41598-019-40233-1). Perhaps adding more PCRs in following studies will increase the species observed? Additionally the number of PCR cycles seems quite high to me. I try to stay below 40 cycles for both PCR steps, here you have 53! See figure 3a of Nichols et al. (2018, doi.org/10.1111/1755-0998.12895) to see the effects of many PCR cycles on the relative proportions.

6. PLOS authors have the option to publish the peer review history of their article (what does this mean?). If published, this will include your full peer review and any attached files.

Reviewer #1: No

---

## [Author Response · Author response to Decision Letter 0]

30 Dec 2020

Response to Reviewers

We have updated the formatting of the manuscript to follow the style requirements for PLOS ONE.

2. In your Methods section, please provide additional location information of the sampling sites, including geographic coordinates for the data set if available.

Geographic coordinates for sampling sites were added to the methods section starting at line 125.

3. We note that you are reporting an analysis of a microarray, next-generation sequencing, or deep sequencing data set. PLOS requires that authors comply with field-specific standards for preparation, recording, and deposition of data in repositories appropriate to their field. Please upload these data to a stable, public repository (such as ArrayExpress, Gene Expression Omnibus (GEO), DNA Data Bank of Japan (DDBJ), NCBI GenBank, NCBI Sequence Read Archive, or EMBL Nucleotide Sequence Database (ENA)). In your revised cover letter, please provide the relevant accession numbers that may be used to access these data. For a full list of recommended repositories, see http://journals.plos.org/plosone/s/data-availability#loc-omics or http://journals.plos.org/plosone/s/data-availability#loc-sequencing.

We uploaded data to NCBI Sequence Read Archive (https://www.ncbi.nlm.nih.gov/sra/PRJNA681428) in addition to making the sequence data available on Dryad.

4. We note that Figure 1 in your submission contain map images which may be copyrighted. All PLOS content is published under the Creative Commons Attribution License (CC BY 4.0), which means that the manuscript, images, and Supporting Information files will be freely available online, and any third party is permitted to access, download, copy, distribute, and use these materials in any way, even commercially, with proper attribution. For these reasons, we cannot publish previously copyrighted maps or satellite images created using proprietary data, such as Google software (Google Maps, Street View, and Earth). For more information, see our copyright guidelines: http://journals.plos.org/plosone/s/licenses-and-copyright.

(1) You may seek permission from the original copyright holder of Figure 1 to publish the content specifically under the CC BY 4.0 license.

The map was created using the free and open source qGIS3 software version 3.0. The base map was made using the Esri Gray Light layer which is has a CC by 4.0 license already. The link to the license is here: https://www.arcgis.com/home/item.html?id=33ea4550c8144e66847d902e4766c2f7 & https://creativecommons.org/licenses/by/4.0/ . On line 134, in the caption of the figure, we included the following language: “The map was generated using the free and open source software QGIS version 3.0”.

We have updated the Supporting Information file naming conventions and included captions to meet the PLOS ONE style guidelines.

Comments to the Author

Reviewer #1: Gold and co-authors present a study comparing eDNA metabarcoding and SCUBA surveys to monitor fish populations in a marine protected area (MPA). The manuscript is well written, technically sound and adds to our knowledge comparing SCUBA and eDNA metabarcoding for applied monitoring of MPAs. The authors show similar species detected by both methods and demonstrate the eDNA metabarcoding can resolve geographically separated sites. However, the authors conclude that the greater estimated species richness outside of the MPA (contrary to expectations and SCUBA observations) detected by eDNA metabarcoding can be attributed to a swamping effect of MPA inhabiting fish. While this might well be true the authors present no data to support the claim. If the authors can provide more support for this argument, or provide a more broad explanation for the observations, then I would support publication of the manuscript. I provide an expansion of my major objection and some minor line specific comments below.

The observation of greater species richness in sites outside of the MPA has three possible explanations in my view. First, there might actually be greater species richness outside of the MPA and eDNA provides evidence for biases in the SCUBA surveys. Second, the sampling design is picking up highly localised species richness differences and including more than one site per site type may reveal the expected pattern. Or third, some methodological bias due to eDNA transport or in the workflow has resulted in an incorrect estimate of species richness. In lines 444-470 the authors seem to think the third option is the most likely, particularly focussing on the effect of eDNA transport in their discussion. I have detailed in a line specific comment below my objection to read depth of samples being listed as a possible explanation. This withstanding I think more broad overview of the possible explanations is required. We simply don’t have enough data to attribute the observations of greater species richness to loss of species detections in the MPA due to a swamping of the PCR with kelp taxa DNA.

New Lines 455-504: We have address the reviewer’s concern with the above section of the discussion and elaborated on additional potential drivers of observed differences species richness patterns between eDNA and SCUBA surveys. Here is the excerpt:

Traditional visual surveys most often report higher biodiversity and biomass inside MPAs [56], including Scorpion State Marine Reserve [7]. However, our results surprisingly indicate lower diversity inside the MPA. This paradoxical result is partially explained by the inability of eDNA to resolve Sebastes species that were visually observed inside (n=3) and on the edge of the MPA (n=1), but not outside. However, Sebastes only accounts for some of the differences inside and outside of the MPA, suggesting that other explanations are required.

One potential source of error could be sampling design of visual and eDNA methods. Time limited SCUBA surveys may not capture species richness as well as eDNA surveys outside the MPA where fish abundance is lower, reducing detection probabilities. Similarly, the nine eDNA samples taken in each region may not capture true patterns of species richness (but see S1 Fig). Distinguishing among these possibilities may be possible by using visual survey protocols that increase transect replication and taxonomic focus [57,58] and using eDNA protocols with increased spatial sampling replication to mitigate patchiness of eDNA dispersion across a reef [9] or by increasing technical PCR replication to reach full species saturation [20].

While the above highlights the difficulty of comparing two surveying methods with imperfect detection rates [55,59], there may be a simpler explanation for eDNA capturing more diversity outside the MPA. Inside an MPA where resident kelp forest fishes are very abundant, the concentration of local fish eDNA is likely very high. In contrast, for transient species passing through these habitats, or for eDNA advected from adjacent habitats, eDNA concentrations would likely be relatively low. However, outside the MPA, where local kelp forest fish are less abundant, both local and transient/advected eDNA concentrations would be low. Because PCR is a probabilistic process, the strongly skewed concentrations of local kelp forest taxa eDNA inside the MPA may reduce the probability of amplifying and detecting rare taxa, local or transient. In contrast, outside the MPA where all eDNA signals are low, the probability of detecting transient and/or advected eDNA would increase.

Support from this hypothesis comes from examining taxa recovered outside the MPA that were not detected inside the MPA. In total, 38.5% of taxa detected outside the MPA were non-rocky reef species such as Yellowtail amberjack (Seriola lalandi, Carangidae), California clingfish (Gobiesox rhessodon, Gobiesocidae), and Fantail flounder (Xystreurys liolepis, Paralichthyidae). These species occasionally pass through rocky reef environments, but their eDNA could also be transported from nearby pelagic, intertidal, and sandy bottom ecosystems [17,43]. In either case, eDNA concentrations would be relatively low, with low detection probabilities inside the MPA but comparatively higher detection probabilities outside. In addition, the MPA site had high kelp abundance while kelp was largely absent outside of MPA (J. Sprague per. obs., 2020). Because kelp creates a three dimensional structure that dampens cross reef flow [60], advection of foreign eDNA should be more likely outside the MPA, potentially increasing the probability of detecting non-local eDNA signatures.

Whatever the cause, the paradoxical pattern of species richness observed in this study highlights that eDNA data must be interpreted judiciously [61]. Metabarcoding methods often perform unexpectedly when DNA concentrations are low, increasing the probability of sequencing rare species [9]. Thus additional ecological metrics to species richness, ones that are more representative of ecological patterns and processes, are needed to optimally interpret eDNA results [16]. These results ultimately highlight the value of ground truthing eDNA results with visual surveys in novel applications to ensure proper interpretation of results [55]. However, they also highlight that comparing survey methods with imperfect detection efforts in the field is challenging as there is no “gold standard” for ecosystem biomonitoring [55,59].

Line 63-68: These observations are great but can we generalise these issues to all SCUBA-based monitoring programs?

New Lines 58-62: The authors David H. Kushner and Joshua Sprague, who manage and operate the Kelp Forest Monitoring Program, are confident the labor estimates are accurate reflections of similar large scale SCUBA based monitoring programs used to monitor nearshore reefs globally. However, upon further deliberation we determined that the cost estimates for the Channel Islands National Park Service Kelp Forest monitoring programs are difficult to calculate and decided to remove this information directly from the text. Instead, in the discussion we cite a few recent cost estimates of eDNA and discuss the difficulties of estimating costs for these very different sampling methods New Lines 365-376:

While eDNA found significant differences inside and outside of the MPA and provided data on more taxa than visual survey methods, a key question remains as whether eDNA provides equivalent biomonitoring data for lower cost and effort [44,45]. We note that estimating the exact costs of visual surveys and eDNA surveys is challenging, given the stark differences in equipment, training, and infrastructure required to support these different biomonitoring efforts [46]. Although in this case we found eDNA was less expensive (Tables S8-9), we note this may not always be the case depending on the locations and species surveyed as well as the expertise of the individuals involved [10]. Regardless, what is important is that eDNA allows individuals with no dive experience or taxonomic training to obtain comparable, and at times, more complete biomonitoring results than from conducting SCUBA-based visual surveys [9,47–49]. Furthermore, this opens up sampling to times and locations where visual surveys cannot be conducted [10]. 

Line 72: Perhaps “…largely limited to…” rather than ““…largely limited o…”

 New Line 69: We have fixed the typo and adopted the reviewer’s suggestion here.

Line 155: I think the Anacapa Toolkit is a nice tool but it relies on loads of software packages that are not cited here. The authors should cite these packages to provide proper attribution to their authors, and also describe the version number of the toolkit and associated underlying software for reproducibility.

New Lines 156-178: We followed reviewers advise and included a description and citation of the packages and version numbers of software underlying the Anacapa toolkit.

Line 159: Should the reference here be a number? If so I think it is number 29?

We have updated the citation formatting throughout.

Line 167: In your data availability statement you mention that the code will be available on GitHub. Please upload this data before submitting the paper – it’s really valuable to be able to scan through code to follow a manuscript. I have received some excellent suggestions from reviewers on my own code and it shows you aren’t afraid for someone to look under the hood of your analysis!

We have now made the code and underlying data needed to run the analyses available on GitHub: https://github.com/zjgold/Scorpion-SMR-eDNA-Metabarcoding . We thank the reviewer for this important suggestion.

Line 169: In my mind a Levine’s test examines equality of variance between groups? I think “homogeneity of dispersions” is also technically correct but this term applies more frequently when testing for between multivariate groups after/before a PERMANOVA procedure.

New Line 177: We have addressed the above comment and updated the language to the following: “subsequent Levine’s test for equality of variance”

Line 180: This function is from vegan. Please cite vegan here or make it clear, please detail the version number of vegan and R used.

We have included the version numbers and package attribution to the software used to conduct the analyses.

Line 261: What does this diagram look like when you use a binary index of species incidence like Jaccard? Does the eDNA index used to generate Bray-Curtis dissimilarity add lots of information to the separation of these samples?

We included eDNA results generated when using a the Jaccard-binary similarity distances in S2 Appendix and Figure S2. Overall we found highly concordant results when using the Jaccard-binary and Bray-Curtis similarity indices suggesting that the Bray-Curtis dissimilarity metric is not adding substantial information to the separation of these samples. 

Lines 449-452: This section is confusing to a reader. Does read depth matter here if species richness curves have saturated? If the authors believe they do then this is a problem that should be discussed further and potentially there should be some analysis that looks at the number of detections per read (or rarefied richness estimates explored). Personally I think the authors provide good evidence that they have sufficient read depth and adding this methodological caveat makes a reader begin to distrust the data unnecessarily.

New Line 462-470: We have followed the reviewer’s advice and rephrased the section to the following: “One potential source of error could be sampling design of visual and eDNA methods. Time limited SCUBA surveys may not capture species richness as well as eDNA surveys outside the MPA where fish abundance is lower, reducing detection probabilities. Similarly, the nine eDNA samples taken in each region may not capture true patterns of species richness (but see S1 Fig). Distinguishing among these possibilities may be possible by using visual survey protocols that increase transect replication and taxonomic focus [57,58] and using eDNA protocols with increased spatial sampling replication to mitigate patchiness of eDNA dispersion across a reef [9] or by increasing technical PCR replication to reach full species saturation [20].”

Line 514: Please also upload the raw sequence data to a public repository (eg NCBI or ENA)

We have uploaded data to NCBI Sequence Read Archive (https://www.ncbi.nlm.nih.gov/sra/PRJNA681428) in addition to making the sequence data available on Dryad.

Supplementary Methods: Over eight PCR reactions per sample are required to saturate OTU detection (Doi et al. 2019, doi.org/10.1038/s41598-019-40233-1). Perhaps adding more PCRs in following studies will increase the species observed? Additionally the number of PCR cycles seems quite high to me. I try to stay below 40 cycles for both PCR steps, here you have 53! See figure 3a of Nichols et al. (2018, doi.org/10.1111/1755-0998.12895) to see the effects of many PCR cycles on the relative proportions.

We thank the reviewer for their insightful feedback on these topics. We included a brief discussion of the need for additional PCR replicates on New lines: 466-470 to potentially explain the under sampling of fish diversity using eDNA. Although we can’t change how many PCR replicates we used for this study, we will strongly consider reducing our PCR amplification steps in the future and appreciate the shared advice and literature.

---

## [Editor Report · Decision Letter 1]

4 Jan 2021

eDNA metabarcoding as a biomonitoring tool for marine protected areas

PONE-D-20-25777R1

Dear Dr. Gold,

We’re pleased to inform you that your manuscript has been judged scientifically suitable for publication and will be formally accepted for publication once it meets all outstanding technical requirements.

Kind regards,

Andrea Belgrano, Ph.D.

Academic Editor

PLOS ONE

Additional Editor Comments (optional):

Please make sure that the link provided for downloading the FishCARD Reference Database is correct since I was not able to access the data.

---

## [Editor Report · Acceptance letter]

29 Jan 2021

PONE-D-20-25777R1 

eDNA metabarcoding as a biomonitoring tool for marine protected areas 

Dear Dr. Gold:

I'm pleased to inform you that your manuscript has been deemed suitable for publication in PLOS ONE. Congratulations! Your manuscript is now with our production department. 

Kind regards, 

on behalf of

Dr. Andrea Belgrano 

Academic Editor

PLOS ONE